# Low-Grade Inflammation in the Pathogenesis of Osteoarthritis: Cellular and Molecular Mechanisms and Strategies for Future Therapeutic Intervention

**DOI:** 10.3390/biomedicines10051109

**Published:** 2022-05-10

**Authors:** M Alaa Terkawi, Taku Ebata, Shunichi Yokota, Daisuke Takahashi, Tsutomu Endo, Gen Matsumae, Tomohiro Shimizu, Ken Kadoya, Norimasa Iwasaki

**Affiliations:** Department of Orthopedic Surgery, Faculty of Medicine and Graduate School of Medicine, Hokkaido University, Kita-15, Nish-7, Kita-ku, Sapporo 060-8638, Japan; taku.e.19861210@gmail.com (T.E.); falcon8863@gmail.com (S.Y.); rainbow-quest@pop02.odn.ne.jp (D.T.); m000053a@yahoo.co.jp (T.E.); gen_matsu_mae@yahoo.co.jp (G.M.); simitom@wg8.so-net.ne.jp (T.S.); kadoya@med.hokudai.ac.jp (K.K.); niwasaki@med.hokudai.ac.jp (N.I.)

**Keywords:** OA, cartilage, low-grade inflammation, DAMPs

## Abstract

Osteoarthritis (OA) is a musculoskeletal disease characterized by cartilage degeneration and stiffness, with chronic pain in the affected joint. It has been proposed that OA progression is associated with the development of low-grade inflammation (LGI) in the joint. In support of this principle, LGI is now recognized as the major contributor to the pathogenesis of obesity, aging, and metabolic syndromes, which have been documented as among the most significant risk factors for developing OA. These discoveries have led to a new definition of the disease, and OA has recently been recognized as a low-grade inflammatory disease of the joint. Damage-associated molecular patterns (DAMPs)/alarmin molecules, the major cellular components that facilitate the interplay between cells in the cartilage and synovium, activate various molecular pathways involved in the initiation and maintenance of LGI in the joint, which, in turn, drives OA progression. A better understanding of the pathological mechanisms initiated by LGI in the joint represents a decisive step toward discovering therapeutic strategies for the treatment of OA. Recent findings and discoveries regarding the involvement of LGI mediated by DAMPs in OA pathogenesis are discussed. Modulating communication between cells in the joint to decrease inflammation represents an attractive approach for the treatment of OA.

## 1. Introduction

Osteoarthritis (OA), the most prevalent form of arthritis, affects 25% of the world’s population, especially those between 55 to 70. With the increasingly aging population, OA is likely to be more prevalent and continue to be the greatest cause of disability, thus decreasing the quality of life and the productivity of people with significant economic burdens on societies and public health services [1,2]. OA is a complex and multifactorial disease that includes contributions from metabolic, epigenetic, genetic, and cellular factors. To date, there is currently no existing disease-modifying therapy, and the current standard of care is limited to pain management and eventual arthroplasty, which is considered to be the only therapeutic strategy for pain relief and for restoring the function of the affected joint. Therefore, the development of a new preservative therapeutics will be needed to prevent the OA process and ultimately restore the activity of people and maintain the productivity of society [3].

OA has been classically considered as wear and tear of articular cartilage degenerative joint disease that can affect any joint in the body, including the knee, hip, spine, and hands. Nonetheless, knee OA is most prevalent and represents the leading musculoskeletal disease that reduces the mobility of individuals [1,2]. Although cartilage damage is a central feature of this disease, an accumulating body of evidence suggests that OA is a whole joint disease since pathological changes frequently seen in OA joints include progressive loss of and destruction of articular cartilage, thickening of the subchondral bone, the formation of osteophytes, inflammation of the synovium (synovitis), degeneration of ligaments, menisci of the knee, and hypertrophy of the joint capsule [4]. However, the emerging evidence for the existence of a link between low-grade inflammation (LGI) and the severity of OA has spurred a new definition of OA, that is, as a low-grade inflammatory disease of the joint [4,5,6]. Importantly, the interplay between cartilage and synovium appears to play the key role in the development of chronic LGI [6,7,8,9].

Cartilage is specialized connective avascular hyaline tissue that covers the surface of bones and functions as a shock absorber. Cartilage is metabolically active and continuously undergoes remodeling, mediated by chondrocytes that constitute the cellular component of cartilage. Chondrocytes are highly specialized cells that reside within the extracellular matrix (ECM) of cartilage and act as sensors of mechanical, ionic, and osmotic signals in the cartilage microenvironment. They regulate matrix synthesis and degradation in response to signals via the secretion of ECM proteins that include type II collagen and proteoglycans and ECM-degrading proteolytic enzymes, including collagenases and aggrecanase. This tightly regulated process is termed cartilage metabolism, essential for maintaining cartilage homeostasis and function [3]. In OA, inappropriate biomechanical and biochemical signals in chondrocytes impair their regulated metabolic activities and shift to catabolic processes where cartilage degeneration exceeds the rate of chondrocyte remodeling due to the increased activity of matrix-degrading enzymes. The continual loss of collagen from ECM with limited ability to repair this loss leads to the death of chondrocytes and the weakening of cartilage structure, thus causing extensive matrix degradation and loss [10,11,12].

The synovium is a layer that seals the synovial cavity and is responsible for maintaining the volume and compositions of synovial fluid, mainly by producing a substance that lubricates and nourishes the cartilage and cells in the joint. In normal conditions, only very low numbers of macrophages are seen in the intimal lining layer and sublining layer of the synovium [13]. Nonetheless, the increasing accumulation of macrophages at the intimal lining layer is the primary morphological feature of synovitis. It is accompanied by an increase in the production of proinflammatory cytokines, such as Interleukin (IL)-1β, IL-6, IL-8, and Tumour Necrosis Factor (TNF)-α, all of which are considered to be the major drivers of ongoing joint destruction and the pathogenesis of OA [8,13,14,15,16]. These findings point to a therapeutic opportunity in which disease-modifying interventions targeting inflammatory processes could be efficacious for the treatment of OA.

The impaired metabolic activity of chondrocytes due to abnormal biomechanical and biochemical signals is associated with the release of degenerated cartilage components and necrotic chondrocytes and their endogenous molecules that act DAMPs in the joint. The ability of DAMPs to activate a broad range of receptors of innate immune cells is the basis of inflammatory response development in the joint. They promote the recruitment and activation of macrophages that produce an array of inflammatory cytokines, thereby facilitating the development of an LGI state in the joint, increasing the synthesis of chondrocyte collagenases and aggrecanase, thereby exaggerating cartilage degeneration [17,18,19,20,21]. Therefore, targeting the interaction between chondrocytes and macrophages that mediate LGI represents a potential therapeutic strategy for the treatment of OA [20]. Given the fact that studying cellular crosstalk between chondrocytes and macrophages is a fundamental step for understanding the mechanism responsible for cartilage degeneration, this review aims to provide a summary of the recent findings highlighting how communication between chondrocytes and macrophages via Damage-associated molecular patterns (DAMPs) can maintain the LGI in the joint microenvironment and initiate the OA process.

## 2. Interplay between Chondrocytes and Macrophages Represents Positive Feedback Loops Triggering Inflammation in the OA Joint

The main components of a healthy joint include cartilage, the synovium, and synovial fluid. Cartilage is a flexible connective tissue that lubricates the contact surfaces between long bones, absorbs stress during movement, and facilitates efficient joint motion. Chondrocytes constitute the cellular component of the cartilage and play an essential role in maintaining tissue homeostasis. In normal metabolic processes, chondrocytes produce ECM components, including type II collagen, glycoproteins, proteoglycans, and hyaluronan, essential for maintaining cartilage structure and function. Importantly, chondrocytes are sensitive to changes in mechanical and biochemical signals in the joint. These factors can have a profound influence on the health and normal function of chondrocytes and the joint [10,22]. Mechanical stress due to hypermobility, overuse, and anatomic misalignment alter the biomechanical signals in the joint resulting in the chronic damage of cartilage that is attributed to the development of meniscal and articular cartilage tears that initiate the OA process. On the other hand, the release of inflammatory cytokines by cells in the joint is associated with increased levels of apoptotic and hypertrophic chondrocytes that propagate the progression of OA. OA involves the breakdown of the balance between the synthesis and destruction of such components and shifts to the catabolic function of chondrocytes that reduce the production of ECM components and the overproduction of ECM-degrading enzymes, including collagenases and aggrecanase, thereby degrading the ECM. Collagenases, namely Matrix Metalloproteinases (MMPs) 1 and 13 are the primary factors leading to the overall degradation of the collagenous framework, while aggrecanases, namely A Disintegrin and Metalloproteinase with Thrombospondin Motifs (ADAMTS) 4 and 5, are the proteolytic enzymes that degrade proteoglycans [4,22]. Loss of cartilage weakens the normal function of the joint concerning gliding movements and the absorbance of mechanical stress, which ultimately results in the development of pain and eventual disability.

The synovium is a thin layer of soft tissue that lines the joint cavity and provides fluid to the joint for efficient movement. In OA, cell infiltration is associated with an increased thickness of this membrane and alterations in synovial fluid components. Synovial fluids that normally contain lubrican and hyaluronic acid facilitate the movement of the joint and provide an appropriate environment for cellular communication in the joint [13]. Macrophages are the predominant cells that infiltrate into the synovial membrane in OA and are believed to be the main source of inflammatory mediators in the joint. Both proinflammatory M1 and anti-inflammatory M2 are present in the synovium of the OA joint; however, an experimental mouse model with proinflammatory M1 bias mice developed greater synovial inflammation and cartilage pathology than mice with an M2 bias [15,23]. The important role of macrophages in cartilage degeneration is experimentally evidenced by the fact that the depletion of macrophages with the intra-articular injection of clodronate-laden liposomes dramatically reduces the production of matrix-degrading enzymes and the breakdown of articular cartilage [24,25]. Synovial macrophages recognize DAMPs, including cartilage fragments and endogenous molecules derived from necrotic chondrocytes that are released into the synovial fluids following tissue damage or cellular stress and initiate inflammatory response. The activation of macrophages in the joint leads to the production of proinflammatory cytokines and MMPs, including IL-1β, IL-6, IL-8, TNF-α, and MMP13, which are considered major drivers of the ongoing joint destruction and the pathogenesis of OA [15,26,27]. The sustained breakdown of cartilage and surrounding tissues provides a continuous source of ECM components and endogenous stimuli that mediate the persistence of the low-grade activation of inflammatory pathways, which maintain the continuing cycle of inflammation and tissue destruction [28,29] (Figure 1).

## 3. DAMPs Switch LGI and the OA Process in the Joint

DAMPs activate Pattern Recognition Receptors (PRRs), including Toll-Like Receptors (TLRs), NOD-Like Receptors (NLRs), and the Receptor for Advanced Glycosylation End products (RAGEs), that are expressed on the surface of macrophages, synoviocytes, and chondrocytes that are present in the joint, and initiate signaling cascades leading to the activation of transcription factors that are involved in the production of inflammatory mediators and matrix-degrading enzymes, thereby progressively destroying cartilage [20,30,31] (Figure 2). Interestingly, the blockade of these receptors has been proven to reduce inflammation in the joint and cartilage degeneration in several experimental OA models, implying that they are involved in the pathogenesis of OA [19,20]. Of these, TLR2/TLR4 signaling which is known to play a role in the recruitment of adapter proteins such as MyD88, TIR domain-containing adaptor-inducing interferon (TRIF), TRIF-related adaptor molecule (TRAM), MyD88-adaptor like (Mal) that activate nuclear factor kappa B (NF-κB), mitogen-activated protein kinase (MAPK) and phosphoinositide 3-kinases (PI3K) pathways resulting in the production of proinflammatory cytokines, Nitric Oxide (NO), Prostaglandin E2 (PGE2) synthesis and MMPs [32]. Importantly, the activation of TLRs in chondrocytes is associated with the downregulation of aggrecans and type II collagen in cells and cartilage [29]. RAGEs are cell-surface receptors of the immunoglobulin superfamily that recognizes DAMPs by their extracellular region and activate the downstream signaling of NF-κB and MAPK pathways leading to the production of proinflammatory cytokines and catabolic factors [33]. NLRs are intracellular sensors of 22 cytoplasmic proteins, including the Nacht domain-containing, Leucine-rich Repeat-containing, and Pyrin domain-containing protein 3 (NLRP3) that mediates the activation of inflammasomes. This leads to the maturation of procaspase-1 and caspases-11/4/5 that facilitate the cleavage of Gasdermin D (GSDMD), which mediates the formation of pores in the cell membrane, thereby causing cell membrane rupture that is associated with the release of IL-1α, β and IL-18 [34]. It is important to note that the complement system represents an additional innate immune mechanism that contributes to the development of inflammation in the OA joint, as evidenced by the fact that high levels of complement components are found in OA synovial fluids compared to healthy ones. It is also evident that ECM components such as decorin and biglycan can also bind complement molecules and activate complement pathways, thus contributing to synovitis and subsequent cartilage degeneration [19,27]. DAMPs can bind and activate a cascade of enzymatic activity of complement molecules resulting in the production of membrane-bound fragments (C3b and C5b) and soluble factors (C3a and C5a) that, in turn, trigger signaling through surface receptors (C3aR, C5aR, and the membrane attack complex; MAC) leading to inflammatory responses and cell lysis (Figure 2).

DAMPs are categorized based on their origin into extracellular and intracellular DAMPs. Extracellular DAMPs originated from the extracellular matrix of cartilage and are released during ECM damage and proteolytic activity in the cartilage (Table 1). The majority of extracellular DAMPs include fibronectin, hyaluronan, biglycan, tenascin c, syndecan-4, and type II collagen and aggrecan fragments that have been reported to be increased in OA synovial fluids after cartilage damage. Fibronectin is an extracellular matrix component known to promote an inflammatory response of macrophages through the activation of TLRs and JNK2, and the p38 MAPK signaling pathway resulting in the production of TNF-α, IL-1β, and IL-8 [35]. Moreover, fibronectin fragments appear to trigger catabolic processes in cartilage by activating chondrocytes to produce proinflammatory cytokines and MMP-1 and -3 [36,37]. Low-molecular-weight hyaluronan, a non-sulfated component of the ECM, has been documented to promote the production of NO and MMP by macrophages via TLR-4 and Myeloid Differentiation Factor 88 (MyD88) signaling [38]. Biglycan is a small leucine-rich proteoglycan that has been documented to increase inflammatory and catabolic states in the joint through binding TLR-2, -4 of macrophages and chondrocytes and in activating Extracellular signal-Regulated Kinase (ERK) and the NF-kB signaling pathway leading to the production of proinflammatory cytokines catabolic factors of cartilages [39,40]. Likewise, tenascin-C is an ECM glycoprotein that acts as a potent activator of TLR4 macrophages and is associated with increased inflammatory responses in the joint [41]. Syndecan-4 (SDC-4) is a cell-surface heparan sulfate proteoglycan that is abundant in inflamed joints due to the damage of cartilage and is known to promote the production of proinflammatory cytokines and catabolic factors in the joint [42]. The type II collagen N-terminal fragment 29-mer fragment and the aggrecan 32-mer fragment derived from damaged ECM can also activate the NF-κB signaling pathway in macrophages and chondrocytes, resulting in the production of proinflammatory and catabolic factors in the joint [43,44,45].

Calcium micro-crystals have been documented to act as DAMPs in the joint, and their levels in synovial fluid are strongly associated with the progression of OA. Calcium-containing crystals, including calcium pyrophosphate dihydrate (CPPD) and basic calcium phosphate (BCP) can interact with both TLRs and NLRs of macrophages resulting in the activation of inflammasomes and the subsequent release of proinflammatory cytokines IL-1β and IL-18. Likewise, such crystals can induce the production of NO and MMPs by chondrocytes in a TLR2-NF-κB signal-dependent manner [46,47].

Intracellular DAMPs are endogenous molecules that act as danger signal molecules (alarmins) and have been implicated in various chronic inflammatory diseases, such as inflammatory bowel disease, atherosclerosis, diabetes, and cancer [48,49,50,51,52]. Alarmins, including the High-mobility Group Box-1 (HMGB1), s100 proteins, Heat Shock Proteins (HSPs), Uric Acid (UA), Adenosine Triphosphate (ATP), IL-1α, IL-33, and Flightless (FliI) are both passively and actively released from cells in response to stress and can bind TLRs and RAGEs) resulting in the activation of transcription factors involved in producing inflammatory mediators and matrix-degrading enzymes. The passive release of these molecules can result from cell necrosis and death, while the active release is mediated by an inflammasome/pyroptosis-dependent mechanism or through secreted extracellular vesicles [19,53,54]. Importantly, high levels of alarmins, such as HMGB1, S100 proteins, IL-1α, ATP, and UA, have been reported to be present in the synovial fluid of OA patients [55,56,57,58,59]. HMGB1 is passively released by necrotic cells or actively secreted by myeloid cells in response to stress and activates TLR-2,-4 and RAGE-mediated ERK and NF-κB signaling, thus resulting in the production of cytokines, chemokines, and MMPs. HMGB-1 can promote the catabolic process via the direct activation of chondrocytes with the production of NO, MMP-3, MMP-13, and ADAMTS-5 [18,19,60,61,62]. Interestingly, HMGB1-neutralizing antibody therapy has exhibited cartilage-protective effects in an experimental OA model by suppressing the catabolic process in chondrocytes [62]. Extracellular S100s, namely S100A8 and S100A9 are secreted by macrophages and chondrocytes and are involved in inflammatory pathways in cartilage via the activation of TLR4 and RAGE. They stimulate the production of inflammatory cytokines and MMPs from macrophages and chondrocytes. The S100s proteins appear to increase the expression of MMPs in chondrocytes in β-catenin-dependent canonical Wnt signaling. Of note, S100A9-deficient mice showed reduced pathological changes in a collagenase-induced OA model, supporting their pathological role in the OA process [63,64]. IL-1α, which is predominantly released by macrophages and chondrocytes in response to stress and inflammasome activation, has been documented to partly contribute to the pathology of OA through promoting the production of NO and MMPs by chondrocytes [19,65]. ATP is another alarmin that is released from cells upon exposure to stress. Extracellular ATP has been shown to stimulate pyroptosis signaling in synovial cells and chondrocytes, resulting in cartilage calcification [66,67]. IL-33 is an endogenous molecule that may exert deleterious effects on joints. Its extracellular form plays an important role in exacerbating inflammatory tissue responses by triggering the MyD88-IRAK-dependent pathway, leading to NF-κB and MAPK activation [68,69]. In support of its pathological function in the joint, the administration of an IL-33 neutralizing antibody was reported to alleviate inflammation in the joint and arthritic development in joint in a collagen-induced arthritis model [70]. Recently, FliI, an intracellular protein that mainly functions as an actin-remodeling protein, and a nuclear receptor co-activator in cells, has been reported to be present in abundance in OA synovial fluids where it acts as an alarmin in cartilage via triggering TLR4-ERK1/2 signaling in chondrocytes leading to the production of chondrocyte MMPs [21].

Furthermore, vascular leaks provide another source of DAMPs that contribute to ongoing inflammation. In fact, plasma proteins such as fibrinogen, Gc-globulin, α1-microglobulin, and α2-macroglobulin have been reported to be elevated in OA synovial fluid secondary to vascular exudation, and their levels are passively correlated with the severity of the disease. Supporting their potential contributory role in OA, fibrinogen, Gc-globulin, α1-microglobulin, and α2-macroglobulin activate macrophages and other innate cells in a TLR4-dependent manner resulting in the production of IL-1β, IL-6, and TNF-α in vitro [19,27].

## 4. Risk Factors Associated with LGI and Joint Destruction in OA

There are a variety of factors that are known to increase the risk of OA, including age, injury, sex, obesity, and various metabolic disorders. Interestingly, there is a strong link between these conditions and the development of systemic LGI, which plays a predominant role in the pathogenesis of these disorders (Figure 3). Systemic LGI contributes to the OA process by initiating a cycle of inflammation in the joint by altering biochemical signals in the joint that ultimately results in impaired metabolic activities of chondrocytes and macrophages, leading to an increased activity of ECM-degrading enzymes and cartilage degeneration [6].

### 4.1. Aging

Aging is the greatest risk factor contributing to the development of OA. Senescent cells due to aging develop into a senescence-associated secretory phenotype (SASP) typified by increased production of proinflammatory agents, catabolic mediators, and DAMPs in the tissue microenvironment [71,72]. The sustained release of these factors leads to the development of a low-grade systemic proinflammatory state, a phenomenon referred to as inflammageing, which is a predominant attribute of aged tissues. Chondrocytes undergo cellular senescence due to mitochondrial dysfunction, oxidative stress, endoplasmic reticulum stress, the accumulation of damaged cellular proteins, and cellular DNA damage due to aging. The accumulation of senescent chondrocytes promotes LGI, which compromises the ability of chondrocytes to maintain cartilage homeostasis and promotes the production of inflammatory cytokines, DAMPs, and ECM-degrading enzymes, leading to cartilage destruction [72,73,74].

### 4.2. Injury

Joint injuries, which at often accompanied by meniscal and ligament tears, joint dislocations, and articular surface fractures, contribute to the eventual development of OA. In some extreme cases, OA progression occurs very rapidly after injury, while others occur decades later after appearing to have healed [75]. However, the progression of OA after injuries is likely dependent on other accompanying risk factors, including age and metabolic disorders. The majority of these injuries occur in sports activities and accidents or activities that involve repetitive, irregular stress on a joint, including kneeling, squatting, lifting loads, and climbing. Although the precise mechanism responsible for OA development after injuries is not currently completely understood, some emerging evidence highlights the involvement of inflammatory responses in the joint in the mechanism. In fact, anterior cruciate ligament (ACL) knee injuries, common in athletes, are often associated with increased production of proinflammatory cytokines in synovial fluids and OA progression [76]. ACL injuries cause abnormal physiologic knee bending with increased contact stress in the posterior medial and lateral compartments under loading, which results in the release of DAMPs in synovial fluids, thereby initiating the inflammatory cycle and exaggerating the OA process [77,78].

### 4.3. Sex

Sex is another risk factor for OA, as women are more susceptible to developing severe OA than men, especially after menopause. In fact, there is a large body of evidence to show that inflammation can increase after postmenopause due to a decreased production of estrogen [79]. Recent findings show a link between inflammation and depression in women with menopause-related symptoms, including hot flashes and physical and sexual changes, as a bridge to developing chronic diseases, including OA [80,81,82]. Another possibility for the increased susceptibility of women to developing OA is that a decrease in the production of estrogen is associated with increased bone remodeling resulting in a thinner subchondral bone that leads to an increased overload of the cartilage [54]. The change in mechanical loading on cartilage would be predicted to increase the possible damage to ECM accompanied by the production of DAMPs from affected chondrocytes and damaged ECM, thereby initiating an inflammatory cycle.

### 4.4. Obesity and Metabolic Disorders

Obesity is also one of the major risk factors that contribute to the development of OA through a weight-dependent pathway that is characterized by excessive joint loading and altered biomechanical patterns, and an increase in both systemic and local inflammation. It was estimated that subjects with a BMI > 30kg/m^2^ are 6.8-times more susceptible to knee OA than other subjects [83,84,85]. Likewise, signs of knee OA have been documented in fat mice fed a high-fat diet for 12 weeks [86]. The increased body weight in obese subjects is associated with an increased biomechanical load in cartilage, thus leading to mechanically induced tissue damage and the activation of mechanotransduction stress signaling in chondrocytes. The production of inflammatory cytokines and DAMPs is triggered, contributing to the cycle of inflammation in the joint [84]. Chronic LGI induced by obesity is thought to originate, at least in part, from the adipose tissues that are expanded due to weight gain that contributes to the production of inflammatory factors and adipokines such as leptin, adiponectin, visfatin, and resistin that are accompanied by the accumulation of proinflammatory macrophages M1 in abdominal adipose tissue [87]. However, there is growing evidence to suggest that proinflammatory macrophages accumulate in synovial joints, and this accumulation appears to be associated with a high level of lipopolysaccharides (LPS) in the body and the synovial fluids in obese patients [88,89].

Other metabolic syndromes that contribute to the progression of OA include hypertension, insulin resistance, hyperglycemia and diabetes, and hypercholesterolemia. Insulin resistance is strongly related to visceral adiposity and increased leptin levels due to the accumulation of subcutaneous fat, which has been implicated in cartilage damage. Hyperglycaemia is associated with the accumulation of glucose that contributes to increased inflammatory response and oxidative stress in joint tissue, thereby exacerbating the pathogenesis of OA [90,91,92]. Moreover, several lines of evidence have pointed to the involvement of cholesterol metabolism in the pathogenesis of OA. In fact, epidemiologic studies have noted a strong link between OA and high serum cholesterol levels [93,94]. A possible mechanism for these conditions is that hypercholesterolemia is associated with increased levels of low-density lipoproteins (LDL) that promote systemic inflammatory response and its accumulation in synovial fluids and chondrocytes seen in OA patients facilitates the development of local inflammatory responses and cartilage degeneration [95,96]. In support of these findings, a high-fat diet induces cartilage degradation via a body weight-independent mechanism, and reducing cholesterol levels significantly alleviates these destructive changes in experimental mouse models [97,98]. These collective discoveries have given rise to the concept of immunometabolism as an attractive target for the treatment of OA. Increasing our knowledge of immunometabolism pathways in OA might provide rational strategies for reversing these pathological changes in the disease [98].

### 4.5. Nutrition and Gut Microbiome Dysbiosis

Recent and ongoing research has highlighted a strong link between the progression of OA and nutrition. In fact, a nutrient surplus and chronic caloric excess not only leads to the expanded accumulation of adipose tissue and increased body weight but also triggers the production of intracellular stress signals that potentiate the production of proinflammatory mediators [99]. It is also noteworthy that consuming significant amounts of saturated fat, polyunsaturated fatty acids, and advanced glycation end products (AGEs) in animal products is associated with systemic inflammation and poses a greater risk of OA progression in individuals [100,101]. Nutrition is involved in gut microbiome dysbiosis, which has recently emerged as a key pathogenic factor in the initiation of and progression of OA. The gut microbiome contributes to chronic local and systemic inflammation. It has been implicated in the initiation of numerous diseases, including rheumatoid arthritis, Crohn’s disease, osteoporosis, sclerosis, Parkinson’s disease, Alzheimer’s disease, and diabetes [102,103,104].

The major bacterial phyla in the healthy gut include Firmicutes and Bacteroidetes, predominantly, *Lactobacillus*, *Bacillus*, *Clostridium*, *Enterococcus*, *Staphylococcus*, *Ruminicoccus, Faecalibacterium*, *Roseburia*, *Dialister*, and *Sphingobacterium* which constitute 90% of the gut microbiome. However, a shift in the microbiome profile toward subdominant bacterial species leads to the development of dysbiosis, associated with excessive porosity of the intestinal barrier leading to the leakage of intestinal contents (leaky gut) and increased endotoxin levels in serum. The leakage of gut microbiota and its products into the circulatory system promotes the chronic activation of the innate immune system and inflammation. It is also noteworthy that an intake of excessive amounts of alcohol and smoking, considered to be risk factors for OA, are associated with an increase in the permeability of the intestinal barrier and altered gut microbiota composition, leading to the development of a chronic inflammation state, thereby exaggerating the OA process. Microbial products such as LPS, peptidoglycan (PGN), and flagellin that have leaked from the gut are recognized by PRRs, TLRs, and NLRs and act as pathogen-associated molecular patterns (PAMPs) or DAMPs that trigger numerous intracellular signaling cascades, leading to the production of inflammatory mediators [105,106]. It is evident that the level of LPS-derived microbiota in serum and synovial fluid is associated with the severity of knee OA [88,89]. It has also been shown that PGNs are involved in the pathogenesis of OA, as they can activate cells in the joint to produce MMPs and proinflammatory cytokines through activating TLR2 and NLRP3 inflammasomes [106,107].

There is a growing body of evidence to indicate that gut microbiota plays a key role in intestinal homeostasis and in regulating the function of the intestinal barrier through producing short-chain fatty acids (SCFAs) and metabolites. Of these SCFAs, butyrate, produced by microbiota via the fermentation of dietary fiber, has been documented to reduce intestinal permeability by promoting tight junctions in epithelial cells. Moreover, butyrate exerts local and systemic anti-inflammatory effects in the intestine and other tissues via promoting the differentiation of CD4 T cells into regulatory T cells (Tregs) and macrophages into an anti-inflammatory phenotype [108]. Consistent with these findings, nutritional supplements of probiotics and prebiotics confer several health benefits in experimental and clinical musculoskeletal disorders models by maintaining intestinal microflora balance, suppressing inflammation and pain, and promoting bone metabolism [109]. In this regard, the administration of *Lactobacillus casei* suppresses synovial inflammation, pannus formation, and articular cartilage and bone degeneration in an arthritis mouse model [110]. Moreover, this treatment has been reported to inhibit inflammation, pain, and cartilage degeneration in experimental OA models [111]. It positively affects bone as it protects ovariectomized mice from bone loss and significantly improves bone mineral density [112]. Consistent with these findings, the pharmaceutical use of probiotics has been documented in multiple clinical studies, and nutritional supplements of probiotics from 6 months have been reported to reduce inflammation and article cartilage degeneration [113]. Moreover, it is also important to note that prebiotic dietary fiber supplementation aids in restoring a healthy microbial community in the gut, resulting in reduced systemic inflammation and improved joint function and integrity [114]. These collective findings point to the importance of nutrition and gut health in preventing the OA process [114,115].

## 5. Dampening LGI as a Strategy for Managing OA

Strategies for preventing cartilage degeneration in OA may require a two-phase therapeutic approach (Figure 4). The first involves controlling the inflammatory state via regulating cellular communication between chondrocytes and macrophages and inhibiting DAMP production and signaling [17,19]. This primarily includes avoiding the overuse and hypermobility of joints that put an excessive load on them, leading to meniscal and cartilage tearing and other compositions of ECM. Importantly, blocking the biological activity of DAMPs derived from chondrocytes and macrophages using a specific ligand and neutralizing antibodies represents a promising therapeutic approach for alleviating the inflammatory and OA processes in the joint. In the same context, an approach aimed at reducing the cellular expression of and the release of DAMPs would be most beneficial for blocking their biological activities. However, it is important to note that although blocking DAMPs receptors, namely PRRs, has shown promising effects in experimental models, these receptors are particularly indispensable in host immune response and defense, and blocking them would likely result in adverse effects such as increased susceptibility to infection [17,19,20]. Neutralizing antibodies to HMGB1, type II collagen peptide Coll2-1, and S100A8/A9 has shown some promising outcomes in that inflammatory effects are inhibited, and the pathological changes associated with arthritis are reduced in rodent models [17,20,62,116,117,118,119]. Likewise, small-molecule inhibitors such as paquinimod and laquinimod derived from quinoline compounds that block the extracellular activity of S100A8/A9 are beneficial for treatment since they were reported to reduce synovial inflammation and cartilage damage in an experimental OA model [19,120,121]. Other compounds that block the secretion of HMGB1 and S100A8/A9, including glycyrrhizin, ethyl pyruvate, and colchicine, have shown pharmacological effects in an experimental OA model [122,123,124]. Importantly, colchicine has been used in clinical trials and showed promise in reducing pain and improving clinical symptoms via inhibiting the secretion of alarmins and the activation of inflammasome signaling [124,125,126]. In the same context, MCC950, a small-molecule chemical inhibitor that selectively inhibits inflammasomes, has been proven to be an effective agent for the treatment of OA and is currently in clinical trials to treat arthritis [127,128]. It should also be noted that the activation and polarization of macrophages represent an additional targetable strategy for the treatment of OA [129]. Therapies that modify the inflammatory state of macrophages and their cytokines, namely anti-TNF agents and IL-1 antagonists, have been reported in experimental models and clinical trials [130,131]. Moreover, methotrexate, which exerts an anti-inflammatory effect by suppressing the inflammatory functions of macrophages, has shown promising clinical outcomes typified by reducing pain and other OA symptoms [132]. Recent findings revealed that extracellular vesicles (EVs) derived from human umbilical cord mesenchymal stem cells ameliorate the progression of OA and exerted chondroprotective potentials in an OA rodent model via promoting M2 macrophages polarization [133]. Likewise, platelet-rich plasma (PRP) has shown the ability to promote M2 macrophage polarization and exerts some promising therapeutic effects for the treatment of OA [134,135,136].

The second approach involves targeting systemic LGI by lifestyle modifications, including weight loss, exercise and nutrition, and dietary supplements. Losing weight appears to be the key approach to managing OA. It helps lower the biomechanical load in cartilage and mechanically induced tissue damage in the joint to reduce subcutaneous fat accumulation and adipose tissue-induced inflammatory responses [109,137,138]. Exercise exerts several benefits on OA as it promotes well-being, mental health, strengthens the muscles, maintains balance and mobility, endurance, and posture, and also can increase the flexibility in affected joints. Exercise promotes muscle secretory functions that can reduce systemic inflammation and cellular senescence and reverse many of the age-related changes in tissues [139,140]. On the other hand, numerous pieces of evidence highlight the existence of a strong link between nutrition and OA development since diet composition plays a key role in the development of LGI [141]. It is evident that diets high in carbohydrates and fat promote systemic inflammation in tissues. Therefore, any pharmacological intervention for OA should also consider nutritional strategies to reduce inflammation, including a healthy diet with probiotics and prebiotics and potent anti-inflammatory/antioxidant supplements. Consuming a low-fat, low-sodium diet with greater intakes of fish and vegetables, including watercress, garlic, onions, parsley, celery, lime, lemon nuts, and seeds, will also have positive effects on the suppression of disease progression [142,143]. The nutritional supplement of probiotics in combination with chondroprotectors, such as glucosamine, can be a choice for the treatment of OA since it can reduce inflammation and regulate cartilage metabolism [109]. Antioxidant supplements, namely omega-3 fatty acids, green tea, polyphenol, blueberry extracts, milk thistle extract (silymarin), stinging nettle extract, and vitamins C and E have also been proven to show beneficial effects in alleviating extrinsic cellular stressors, cellular senescence and systemic inflammation [54,143].

## 6. Conclusions

Emerging evidence underlines that chronic LGI plays a central role in the development of OA through maintaining synovial inflammation and propagating chondrocyte catabolic responses leading to cartilage degeneration. Importantly, the modulation of communication between synovial macrophages and chondrocytes via targeting DAMPs from ECM damage or necrotic cells provides an excellent opportunity for dampening the intensity of inflammation in an OA joint and the subsequent OA process. Probiotics, prebiotics, and nutritional antioxidant supplementations along with exercise should also be considered in the treatment of OA due to their positive influence on the body’s health and for reducing systemic inflammation. However, given that DAMPs are the major stimuli for initiating and maintaining the inflammatory cycle in the joint, further research aimed at a better understanding of the pathological functions of DAMPs in OA is needed and is expected to provide new clues for the discovery of therapeutic targets.

## Figures and Tables

**Figure 1 biomedicines-10-01109-f001:**
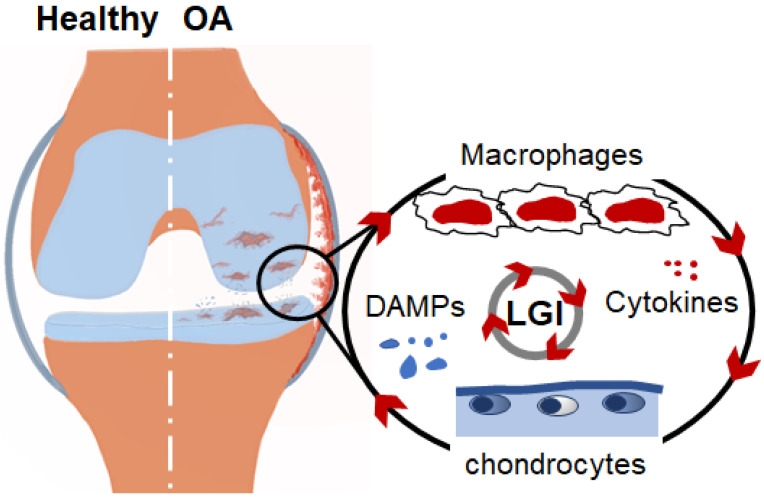
Schematic representation of inflammation cycle in the OA joint. Cartilage damage tissue and tears, crystals, endogenous molecules from apoptotic chondrocytes, and plasma proteins from vascular leak act as DAMPs which trigger local LGI in the joint, thus promoting the production of proteolytic enzymes. Continual release of DAMPs in the joint leads to chronic activation of innate immune cells, including macrophages, thus amplifying a vicious inflammation cycle leading to cartilage degeneration.

**Figure 2 biomedicines-10-01109-f002:**
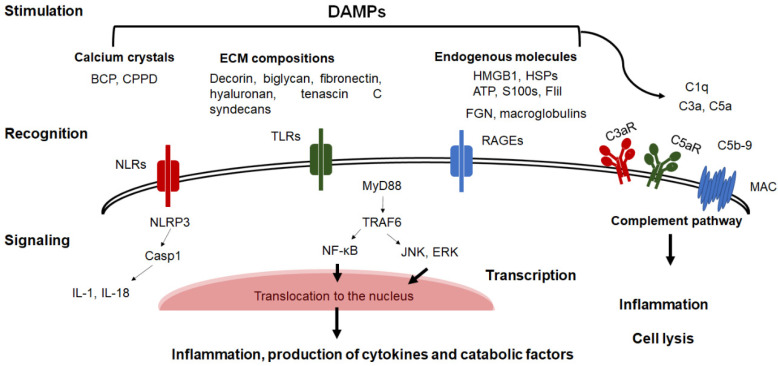
Schematic representation of the role of DAMPs in the initiation of LGI. DAMPs, including cartilage compositions, crystals, endogenous molecules, and plasma proteins, are recognized by PPRs, including NLRs, TLRs, and RAGEs resulting in the activation of several signaling pathways involved in the production of proinflammatory cytokines and proteolytic enzymes by macrophages and chondrocytes. The binding of DAMPs to complement molecules activate the complement pathway resulting in cell activation leading to inflammatory responses and cell lysis.

**Figure 3 biomedicines-10-01109-f003:**
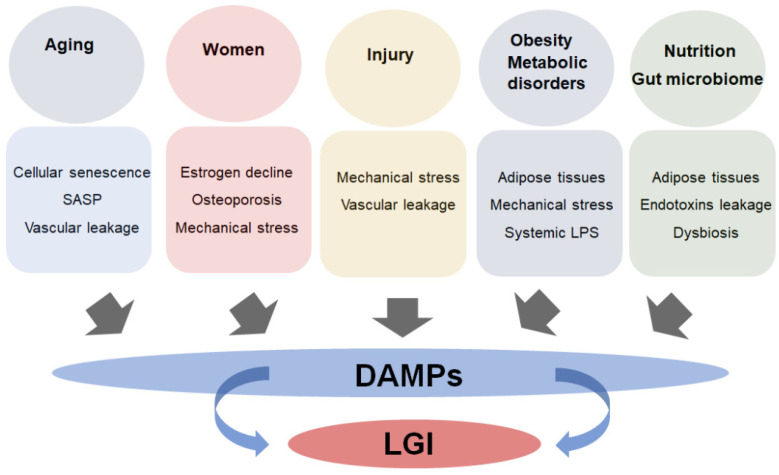
Association between LGI state and the risk factors of OA. Factors including aging, sex (women), injury, obesity and metabolic syndromes, nutrition, and gut microbiome are associated with increased production of DAMPs in the tissues, which facilitate the development of LGI.

**Figure 4 biomedicines-10-01109-f004:**
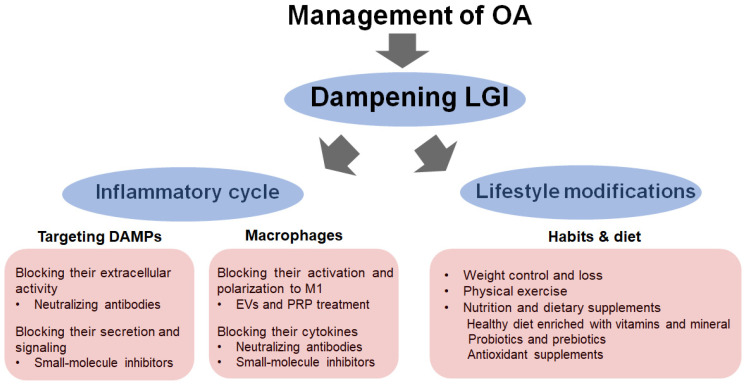
Targeting LGI for treatment of OA. Dampening inflammation by targeting the inflammatory cycle includes blocking the extracellular activity and production of DAMPs, reducing macrophage activation and polarization, and blocking inflammatory cytokines, TNF-α, IL-1, and IL-6. Lifestyle modifications include weight loss, exercise and nutrition, and dietary supplements.

**Table 1 biomedicines-10-01109-t001:** DAMPs and their signaling pathways are implicated in inflammation and OA pathogenesis.

DAMPs	Signaling	References
**ECM compositions**		
Decorin and biglycan	C1q; Complement system	[19,27]
Fibronectin	TLR2, 4/JUK2-MAPK	[35]
Hyaluronan	TLR4/MyD88	[38]
Biglycan	TLR2, 4/ERK/NF-κB	[39,40]
Tenascin c	TLR4/MAPK	[41]
Syndecan-4	MAPK	[42]
Type II collagen 29-mer fragment	TLRs/NF-κB	[43,44]
Aggrecan 32-mer fragment	TLR2/NF-κB	[45]
**Crystals**		
CPPD, BCP	TLR2/NF-κB NLR/NLRP3	[46,47]
**Alarmins**		
HMGB1	TLR2, 4/ERK/NF-κB	[60,62]
s100sA8, S100A9	TLR4/NF-κB	[63,64]
UA	NLR/NLRP3	
ATP	NLR/NLRP3	[66,67]
IL-1α	IL-1R1/MAPK	[65]
IL-33	MyD88/NF-κB/MAPK	[68,69]
FliI	TLR4/ERK1	[21]
**Plasma proteins**		
Fibrinogen	TLR4/NF-κB	[19,27]
Gc-globulin	C5a; Complement system	[19,27]
α1-microglobulin, α2-macroglobulin	MAPK-ERK/NF-κB	[19,27]

## Data Availability

Not applicable.

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
