# Peer review of "Low-Grade Inflammation in the Pathogenesis of Osteoarthritis: Cellular and Molecular Mechanisms and Strategies for Future Therapeutic Intervention"

_biomedicines, 2022, doi:10.3390/biomedicines10051109_

Round 1

Reviewer 1 Report

This is a very comprehensive and well-written review, which may be relevant not only to the specialists in the field but also to the broader readership of the journal.

However, in the therapeutic armament of OA treatment I missed discussion of the use of MSC-derived exosomes as a treatment way, which may relieve OA by promoting the phenotypic transformation of synovial macrophages from M1 to M2. In my opinion, addition of a discussion point on this issue would substantially improve the quality of the manuscript.

Author Response

We thank the reviewer for this assessment and the useful and constructive suggestions made during the first round of review.

Reviewer 1.

This is a very comprehensive and well-written review, which may be relevant not only to the specialists in the field but also to the broader readership of the journal.

However, in the therapeutic armament of OA treatment I missed discussion of the use of MSC-derived exosomes as a treatment way, which may relieve OA by promoting the phenotypic transformation of synovial macrophages from M1 to M2. In my opinion, addition of a discussion point on this issue would substantially improve the quality of the manuscript.

We acknowledge the reviewer comment and we have added a new paragraph for use of MSC-derived exosomes as a treatment way for reducing M1 macrophages in synovium. Please refer to the page 11, lines 472-475.

Reviewer 2 Report

Dear Authors,

I have read the review and I send you my comments:

1) please clarify it is a review? a narrative review a natural review?

2) Section 1 and 2 are very long please reduce them and add 1 table for section and 1 figure for section

3) please add the definition for DAMP, it is reported inn abstract but not in the text, it is also for other words.

4) section 4.4 please add 1 table and 1 figure

5) the title is "Communication between macrophages and chondrocytes in the osteoarthritic joint as a potential target for therapeutic intervention" but only in section 5 you described the treatment. I think that it may be revised completely. 2 sections must be of introduction and 3 sections or more for the  treatment.

6) Regarding the treatment several references are missing as well as the time of treatment, the dosage and the indications (considering to add tables) as references you can read and cite Rheumatol Int. 2019 Sep;39(9):1661-1662. and Osteoarthritis Cartilage. 2013 Sep;21(9):1400-8. doi: 10.1016/j.joca.2013.06.026

Author Response

Reviewer 2.

I have read the review and I send you my comments:

1) please clarify it is a review? a narrative review a natural review?

We first thank for the reviewer for his instructive comments. Our review is comprehensive article covering the recent findings on the pathogenesis of OA. We believe that knowledge in review is relevant not only to the specialists in the field but also to the broader readership.

2) Section 1 and 2 are very long please reduce them and add 1 table for section and 1 figure for section

Thank you for your comment but we feel sections are important as introduction to give the reader the general knowledge about the pathogenesis of OA and the interaction between macrophages and chondrocytes and its link to inflammation. Section 2 has a figure. However, concerning the reviewer comment, we added a new table describing role of DAMPs in development of inflammation.

3) please add the definition for DAMP, it is reported inn abstract but not in the text, it is also for other words.

We acknowledge the reviewer comment and we have added definition for DAMP and LGI in the text.

4) section 4.4 please add 1 table and 1 figure

Thank you for your comment but section 4 has a figure and adding table might be just repetition of the contents of the figure.   

5) the title is "Communication between macrophages and chondrocytes in the osteoarthritic joint as a potential target for therapeutic intervention" but only in section 5 you described the treatment. I think that it may be revised completely. 2 sections must be of introduction and 3 sections or more for the treatment.

We acknowledge the reviewer comment and we have changed the title to reflect the contents of the manuscript. The new title is as follows: “Low-grade inflammation in the pathogenesis of OA: Cellular and molecular mechanisms and strategies for future therapeutic intervention”.

6) Regarding the treatment several references are missing as well as the time of treatment, the dosage and the indications (considering to add tables) as references you can read and cite Rheumatol Int. 2019 Sep;39(9):1661-1662. and Osteoarthritis Cartilage. 2013 Sep;21(9):1400-8. doi: 10.1016/j.joca.2013.06.026

We acknowledge the reviewer comment and we have added new paragraph for treatment and cited the mentioned article. Please refer to the pages 11-12, lines 451-478.

Reviewer 3 Report

I have reviewed the indicated article which discusses the potential relationship between macrophages in a joint such as a knee and the development and progression of osteoarthritis. Clearly macrophages are present in the OA joint and some can be detected in synovial fluid. Likely, in an activated state, some subsets of macrophages could be contributing to joint inflammation and thus, are potential targets for interventions. However, these calls are only one of several potential targets and targeting multiple cells/molecules may be required to control OA development and progression, particularly as the environment transitions from an acute to chronic state. This transitioning should be mentioned/discussed as the ease of interfering with the processes will likely decline when things are chronic vs acute.

The article is well written, with considerable background information presented and documented. Certainly, there is considerable literature that indicates that induction of inflammation via injury to a ligament such as the ACL, surgery to repair such injuries can lead to induction of OA and that interfering with such inflammation early with anti-inflammatories can diminish development of OA-like pathology. Some discussion of this body of information would be beneficial to the current manuscript. Furthermore, it is also clear that the knee is a mechanically active joint and that the effectiveness of inflammatory mediators such as IL-1 on cells is blunted when the target cells or chondrocytes are mechanically activated. Thus, the mere presence of some inflammatory mediators in a mechanically active joint does not mean they are as active as in vitro studies performed in the absence of mechanical loading might indicate. These points such be discussed in the current manuscript to offer a balanced perspective.

Finally, the authors should change the use of the term "gender" throughout the article (gender is a sociologic term) to the term "sex" as what the authors are describing are biological determinants which is sex.

Author Response

Reviewer 3.

I have reviewed the indicated article which discusses the potential relationship between macrophages in a joint such as a knee and the development and progression of osteoarthritis. Clearly macrophages are present in the OA joint and some can be detected in synovial fluid. Likely, in an activated state, some subsets of macrophages could be contributing to joint inflammation and thus, are potential targets for interventions. However, these calls are only one of several potential targets and targeting multiple cells/molecules may be required to control OA development and progression, particularly as the environment transitions from an acute to chronic state. This transitioning should be mentioned/discussed as the ease of interfering with the processes will likely decline when things are chronic vs acute.

We acknowledge the reviewer comment and we have added a new paragraph for acute and chronic OA after joint injury. Please refer to the pages 8-9, lines 309-325.

The article is well written, with considerable background information presented and documented. Certainly, there is considerable literature that indicates that induction of inflammation via injury to a ligament such as the ACL, surgery to repair such injuries can lead to induction of OA and that interfering with such inflammation early with anti-inflammatories can diminish development of OA-like pathology. Some discussion of this body of information would be beneficial to the current manuscript. Furthermore, it is also clear that the knee is a mechanically active joint and that the effectiveness of inflammatory mediators such as IL-1 on cells is blunted when the target cells or chondrocytes are mechanically activated. Thus, the mere presence of some inflammatory mediators in a mechanically active joint does not mean they are as active as in vitro studies performed in the absence of mechanical loading might indicate. These points such be discussed in the current manuscript to offer a balanced perspective.

We acknowledge the reviewer comment and we have added a new paragraph joint injury. Please refer to the pages 8-9, lines 309-325. Figure 3 is modified accordingly.

Finally, the authors should change the use of the term "gender" throughout the article (gender is a sociologic term) to the term "sex" as what the authors are describing are biological determinants which is sex.

We acknowledge the reviewer comment and we have changed the word gender to “sex” throughout the manuscript.

Reviewer 4 Report

The authors of this review have focussed on the effect of damage associated molecular pattersn (DAMPs) on macrophage and chondrocyte interaction. They summarise the different DAMPs and the mechanisms initiated within cartilage during osteoarthritis and ways that can be used to reduce their effects (e.g. dietary changes).

The authors describe in good detail pathways activated within the cells during release of DAMPs. However, the authors should address the following: -

  1. The complement system is described in detail within section 3 of the review. The authors should add additional figure summarising the interactions of the complement system.
  2. Macrophages are described in detail in both introduction, although in later sections, only DAMPs and downstream signalling cascades are described. Can the authors be more explicit on the macrophage interaction in these sections, as opposed to chondrocytes to avoid confusion for the reader ? Additionally, if this cannot be done, I would suggest the title changed to reflect the main section of this review
  3. A summary figure describing strategies to reduce LGI is required to complement the section described.
  4. Can the authors add an additional section on the drugs involved in reducing LGIs and a summary figure to complement this part ? There are drugs used to reduce LGI and these should be described in detail and their interaction with chondrocytes and/or macrophages.

Author Response

Reviewer 4.

The authors of this review have focussed on the effect of damage associated molecular pattersn (DAMPs) on macrophage and chondrocyte interaction. They summarise the different DAMPs and the mechanisms initiated within cartilage during osteoarthritis and ways that can be used to reduce their effects (e.g. dietary changes).

The authors describe in good detail pathways activated within the cells during release of DAMPs. However, the authors should address the following: -

  1. The complement system is described in detail within section 3 of the review. The authors should add additional figure summarising the interactions of the complement system.

We acknowledge the reviewer comment and we have included the complement system in OA joint in the Figure 2. Please refer to the Page 5.

  1. Macrophages are described in detail in both introduction, although in later sections, only DAMPs and downstream signalling cascades are described. Can the authors be more explicit on the macrophage interaction in these sections, as opposed to chondrocytes to avoid confusion for the reader ? Additionally, if this cannot be done, I would suggest the title changed to reflect the main section of this review

We acknowledge the reviewer comment and we have changed the title to reflect the contents of the manuscript. The new title is as follows: “Low-grade inflammation in the pathogenesis of OA: Cellular and molecular mechanisms and strategies for future therapeutic intervention”.

  1. A summary figure describing strategies to reduce LGI is required to complement the section described.

We acknowledge the reviewer comment and we have added a new figure describing strategies to reduce LGI. Please refer to the Figure 4, page 12.

  1. Can the authors add an additional section on the drugs involved in reducing LGIs and a summary figure to complement this part ? There are drugs used to reduce LGI and these should be described in detail and their interaction with chondrocytes and/or macrophages.

We acknowledge the reviewer comment and we have added new paragraph for treatment and cited the mentioned article. Please refer to the pages 11-12, lines 451-478.

Round 2

Reviewer 2 Report

Dear Authors,

I have not further comments

Reviewer 3 Report

I have reviewed the revised version of the indicated manuscript. The authors have made extensive modifications to the original manuscript which has strengthened it considerably.

The new title is much improved and more accurately defines the topics addressed. The new paragraphs that address my original concerns are fine.

Finally, the authors have added a fairly large number of new citations/references to the manuscript which also support their perspective.

I have no further concerns.

Reviewer 4 Report

The authors have answered my questions appropriately